# Near-Isometric Properties of Kronecker-Structured Random Tensor Embeddings

**Qijia Jiang**
Lawrence Berkeley National Laboratory
qjiang@lbl.gov

## Abstract

We give uniform concentration inequality for random tensors acting on rank-1 Kronecker structured signals, which parallels a Gordon-type inequality for this class of tensor structured data. Two variants of the random embedding are considered, where the embedding dimension depends on explicit quantities characterizing the complexity of the signal. As applications of the tools developed herein, we illustrate with examples from signal recovery and optimization.

## 1 Introduction

It is hardly an overstatement to proclaim that underpins most of the analysis for high-dimensional statistics and structured signal recovery is the heavy hammer made possible by the machinery of Gaussian process, and in particular Gordon-type inequality that gives tight characterization of the suprema of the empirical process with geometric properties of the underlying index set. In this paper, we put Kronecker-structured random tensors into scrutiny and ask for analog of Gordon's inequality for correspondingly tensor-structured signals. We embark with a brief reminder of the classics.

### 1.1 Gordon's inequality for Gaussian random matrix

For signal $u \in T \subset \mathbb{R}^n$ a vector, it is known for $S \in \mathbb{R}^{m \times n}$ random i.i.d standard Gaussian matrix,

$$\mathbb{E}[\min_{u \in T} \|Su\|] \geq a_m - w(T) \quad \text{and} \quad \mathbb{E}[\max_{u \in T} \|Su\|] \leq a_m + w(T)$$

for $a_m = \mathbb{E}[\|g_m\|] \approx \sqrt{m}$ where $g_m \sim \mathcal{N}(0, I_m)$ and $w(T) = \mathbb{E}[\max_{x \in T} g^\top x]$ the Gaussian width for set $T \subset \mathbb{S}^{n-1}$, a subset of the unit sphere. This statement hinges on the Gaussian min-max comparison lemma (i.e., Fernique-Slepian theorem), which implies for $g, h$ independent standard Gaussian vectors,

$$\mathbb{E}_{g,h}[\min_{u \in T} \max_{v \in \mathbb{S}^{m-1}} g^\top v + h^\top u] \leq \mathbb{E}_S[\min_{u \in T} \max_{v \in \mathbb{S}^{m-1}} v^\top Su]. \tag{1}$$

This trades the quadratic form for a more innocuous separable process, from which one can see that the LHS evaluates to the first part of the previous display. The other side is essentially similar. For this expectation bound to justify the attention it deserves, one needs to recognize that $\min_{u \in T} \|Su\|$ (analogously for max) is a Lipschitz function in the Gaussian random matrix $S$, from which (dimension-free) concentration inequality, alongside the bound on the expectation derived above, conspire to deliver a uniform concentration bound as stated below.

**Theorem 1** (Gordon's escape through the mesh [12])**.** *For all $u \in T \subset \mathbb{R}^n$, where $T$ is a (not necessarily convex) cone, with probability at least $1 - 2\exp(-\delta^2/2)$ for $S$ entrywise i.i.d standard Gaussian,*

$$(1 - \epsilon)\|u\| \leq \frac{1}{a_m}\|Su\| \leq (1 + \epsilon)\|u\|$$

*when $m \geq \frac{(w(T)+\delta)^2}{\epsilon^2}$.*

36th Conference on Neural Information Processing Systems (NeurIPS 2022).

Later work of CGMT [17] showed that the reduction of (1) is essentially tight for convex sets, which has surprising consequences for analyzing the risk of various statistical estimators in a high-dimensional asymptotic regime. This elegant analysis, nevertheless, cannot be carried out beyond the Gaussian case due to the lack of comparison lemma (1) (even for subgaussian), but gives that for example, the extreme singular values of a Gaussian random matrix $1/\sqrt{m} \cdot S$ scales as $1 \pm \sqrt{n/m}$ by picking $T = \mathbb{S}^{n-1}$. It also recovers the familiar Johnson-Lindenstrauss lemma for distance-preserving random projection of finite point set when $w = \sqrt{\log(|T|)}$, where $|T|$ is the cardinality of the set.

Seemingly a natural obsession for probabilists for its mathematical allure, results of this flavor have found unexpectedly number of applications across many areas in numerical linear algebra, signal processing, theoretical computer science, among others. Such uniform convergence result is frequently encountered for deriving tight sample complexity bounds for recovery problems, where the problem boils down to characterizing the probability that a random subspace (i.e., null space of Gaussian measurement matrix) distributed uniformly misses the tangent cone of a regularizer. Nonconvex gradient-based optimization heavily leans on these tools for characterizing restricted singular value for deriving convergence with Empirical Risk Minimization. Sketching-based least-squares optimization $\min_x \|SAx - Sb\|_2^2$ also crucially rely on such results, where $w(U \cap \mathbb{S}^{n-1}) = \sqrt{\dim(U)}$ for $U = \mathrm{colspan}([A, b])$ for the subspace embedding property.

## 1.2 Contributions

We aim to generalize Gordon's uniform concentration result for tensor-structured signal $x = u^1 \otimes \cdots \otimes u^d$ while insisting on efficient computation of the embedding operation. More concretely, we consider Kronecker-structured random rank-1 tensor, which when acting on rank-1 tensor-structured signals, can be performed without explicitly forming the $n \times n \times \cdots \times n$ tensor since it can be done factor-by-factor effortlessly. Formally we set out our roadmap to address the following questions:

1. For (1) structured and fast tensored embedding (e.g., Tensor-SRHT as defined in Definition 1 below); and (2) Tensor-Subgaussian introduced in Definition 2, what is dictated from the embedding dimension $m$ for the following guarantee to hold w.h.p

$$\left| \frac{1}{m} \sum_{i=1}^{m} \prod_{j=1}^{d} \langle v_i^j, u^j \rangle^2 - \|x\|^2 \right| \leq \max(\epsilon, \epsilon^2) \cdot \|x\|^2, \tag{2}$$

   for all $x = u^1 \otimes \cdots \otimes u^d \in T^1 \times \cdots \times T^d$ (Cartesian product of $d$ not necessarily convex cones), as a function of the geometric properties of the *individual* sets $T^1, \cdots, T^d$. This is a generalization of the Restricted Isometry Property (RIP) to (1) higher order tensored signals; (2) general cones beyond sparsity. Both sketches above are row-wise tensored and take the form $S_i = \mathrm{vec}(v_i^1 \otimes \cdots \otimes v_i^d)$ for each row $i \in [m]$. We are interested in the regime $m \ll n^d$ and instantiate the embedding result for this sketch from Section 4 to bound the restricted singular value as required by a tensor signal recovery problem in Section 6.1.

2. To improve the dependence of $m$ on the degree $d$ (while maintaining computation efficiency), we consider a recursive embedding in Section 5 which repeatedly calls a degree-2 Tensor-SRHT $S^j \in \mathbb{R}^{m \times nm}$ as a subroutine as follows: $S(u^1 \otimes u^2 \otimes u^3 \cdots) := S^1(u^1 \otimes S^2(u^2 \otimes S^3(u^3 \otimes \cdots)))$. Similar uniform concentration is derived on the scaling of $m$ with geometric properties of the individual sets for this alternative embedding, which is in turn called upon to speed up solving for optimization problem in Section 6.2.

3. Our technique is based on generic chaining - we include comparison with results one would get from more naive method in Section 3 and part with some discussions of lower bound on the embedding dimension in Section F and numerical results in Section 7.

We pause to emphasize it is the correlation in the tensor structure that introduces difficulty for tight concentration – result for general random tensor with i.i.d entries is less challenging to obtain, but at the same time less efficient to apply.

**Definition 1** (Tensor-SRHT). *A random matrix constructed as $S = \frac{1}{\sqrt{m}} P_1 H_n D_1 \circ \cdots \circ P_d H_n D_d \in$*

$\mathbb{R}^{m \times n^d}$ *is called a Tensor-SRHT (Subsampled Randomized Hadamard Transform), if when acting on a rank-1 degree-$d$ tensor, takes the form $S(u^1 \otimes \cdots \otimes u^d) = \frac{1}{\sqrt{m}} P' H_{n^d} D' vec(u^1 \otimes \cdots \otimes u^d) :=$*

$\frac{1}{\sqrt{m}}P_1 H_n D_1 u^1 \odot \cdots \odot P_d H_n D_d u^d$, where $D'$ is a $n^d \times n^d$ diagonal matrix with entries $D_1 \otimes \cdots \otimes D_d$ (i.e., tensor product of independent Rademachers) and $P'$ is a $m \times n^d$ subsampling matrix with a single $1$ in each (independent) row and $H_{n^d} = H_n \otimes \cdots \otimes H_n$ where $n$ is a power of $2$ is the Hadamard matrix of size $n^d \times n^d$. Here $\odot$ denotes Hadamard product and $\circ$ denotes the transposed Khatri-Rao product. Moreover, such embedding can be carried out in time $\mathcal{O}(d(n \log n + m))$.

**Definition 2** (Tensor-Subgaussian). *We call $S \in \mathbb{R}^{m \times n^d}$ a Tensor-Subgaussian embedding if every row $S_i = vec(v_i^1 \otimes \cdots \otimes v_i^d)$ is constructed where each factor is an independent $\sigma$-subgaussian isotropic random vector, i.e., (1) $\mathbb{E}[\langle v_i^j, u^j \rangle^2] = \|u^j\|_2^2$; (2) $\mathbb{E}[|\langle v_i^j, u^j \rangle|^p]^{1/p} \leq \sqrt{\sigma p} \|u^j\|_2$ for all $p \geq 2$, $i \in [m], j \in [d]$ and any $u^j \in \mathbb{R}^n$.*

## 2 Related Work

In the case of vector-valued signal ($d = 1$), embedding analysis for infinite sets using structured matrices requires ingenuity and is significantly more involved in general. Notable extensions include [5, 10]. The work of [15] offered a unifying theme - the important message behind is that one can have a reduction from RIP based result to Gordon-type inequality by invoking it at different sparsity levels with various distortions à la Talagrand's multi-resolution generic chaining. An orthogonal thread for generalizing to heavier-tail distribution involves small-ball technique which gives an one-sided bound for nonnegative empirical process - such undertaking is present in e.g., [18].

Previous work on tensor concentration are mostly concerned with operator norm bounds for symmetric subgaussian and/or log-concave (potentially non-isotropic) factors [11, 23], where for symmetric forms $\|S\|_{op}$ is maximized by a single vector $u \in \mathbb{S}^{n-1}$ therefore for this we only need to content ourselves with a single index set and look at moment deviations of type: $\sup_{u \in \mathbb{S}^{n-1}} \left| \frac{1}{m} \sum_{i=1}^{m} \langle S_i, u \rangle^d - \mathbb{E}[\langle S, u \rangle^d] \right|$, an arguably simpler task. Indeed, a multi-resolution approach is not strictly beneficial here compared to more elementary arguments [11].

The case of non-symmetric factors warrant more care. Both [21, 4] studied pointwise tail bound of the form $\mathbb{P}(|\|Sx\|_2 - \|S\|_F| \geq t)$ for $S \in \mathbb{R}^{m \times n^d}$ a linear mapping, $x = u^1 \otimes \cdots \otimes u^d \in \mathbb{R}^{n^d}$, where $u^k$'s are independent factors each with independent, mean 0, unit variance, subgaussian coordinates – this can in turn be used for deriving a high-probability lower bound on $\sigma_{\min}(X)$ for the $n^d \times m$ random matrix $X$ where each column is formed by the aforementioned tensor $x$. Uniform deviation for general sets on tensors can be viewed as a special instance of 2nd-order chaos with mixed tails [16]. For example in the case of processes with subgaussian-subexponential increments (as is the case when $d = 2$ for Tensor-Subgaussian embedding in Definition 2), i.e., $\forall u > 0, s, t \in T$,

$$\mathbb{P}(\|X_t - X_s\| \geq \sqrt{u} d_2(t, s) + u d_1(t, s)) \leq 2e^{-u},$$

the result of [9] gave a uniform deviation for $\sup_{t \in T} \|X_t\|$ as a combination of $\gamma_2(T, d_2)$ and $\gamma_1(T, d_1)$ but crucially these quantities are tied to the metric complexity of the *product index set* $T := T^1 \times T^2$ – something that is hard to compute by and large. Various works also study finite set embedding for Kronecker-structured sketches, some of which we will leverage for our results and will mention them in later contexts.

## 3 Discrete JL and a Single-scale Approach

At the heart of the following result is a generalized Khinchine inequality [2] which says if $\mathbb{E}[|\langle v^k, a \rangle|^p]^{1/p} \leq C_p \|a\|_2$ for any vector $a \in \mathbb{R}^n$ and all independent $\{v^k\}_{k=1}^d$, then $\mathbb{E}[|\langle v^1 \otimes \cdots \otimes v^d, a \rangle|^p]^{1/p} \leq C_p^d \|a\|_2$ for any (not necessarily rank-1) tensor $a \in \mathbb{R}^{n^d}$. This is closely related to an earlier result from [13] on the concentration of Gaussian chaos but generalized to broader class. Such moment control is only a hop away from tail bounds using standard arguments. We establish the finite-set embedding property for the row-wise-tensored embedding matrices below, building upon previous work. This serves as the stepping stone for the embedding of general sets.

**Lemma 1** (Discrete-JL property for Tensor-SRHT and Tensor-Subgaussian). *For a set of cardinality $p$ that the rank-1 tensor $x \in \mathbb{R}^{n^d}$ belongs, with probability at least $1 - e^{-\eta}$ for any $\eta > 0$ and $\epsilon > 0$, Tensor-SRHT as defined in Definition 1 satisfies $|\|Sx\|_2^2 - \|x\|_2^2| \leq \max(\epsilon, \epsilon^2)\|x\|_2^2$ simultaneously for all $p$ points provided $m = \mathcal{O}(C^d \epsilon^{-2}(\log^d(p) + (1 + \eta)^d))$. The same guarantee holds for*

*Tensor-Subgaussian in Definition 2 with $m = \mathcal{O}(C^d \sigma^{2d} \epsilon^{-2}(\log^d(p) + (1+\eta)^d))$ for some universal constant $C$.*

*Remark.* Close inspection of the proof for Theorem 3 in [2] in fact uncovers that the discrete JL property above holds for more general class of SORS (Subsampled Orthogonal Random Sign) constructions for which $H^*H = n \cdot I_n$ and $\max_{i,j \in [n]} |H_{ij}| \leq c$. In the case $d = 1$, it also recovers the classical Johnson–Lindenstrauss lemma.

Without taking the multi-scale route, in the case $d = 1$, to guarantee $\epsilon$-distortion over a continuous set, one needs to roughly speaking build a $\Delta$-net for $x \in \mathbb{R}^n$ for $\Delta \lesssim \epsilon \cdot \sqrt{m/n}$ therefore the sample complexity one gets with a single-scale approach will scale as

$$ m \gtrsim \frac{\log(|\mathcal{N}^\Delta|)}{\epsilon^2} \gtrsim \frac{nw^2(T)}{m\epsilon^4} \Rightarrow m \gtrsim \frac{\sqrt{n}w(T)}{\epsilon^2}, $$

where we used Sudakov's minorization for bounding the size of the covering with Gaussian width of the set and the JL Lemma for SRHT/Subgaussian matrices for the first transition. This back-of-the-envelope calculation showcases that uniform covering is far from optimal, since in general it could be the case $w(T) \ll \sqrt{n}$ for $T \subset \mathbb{S}^{n-1}$ a subset of the unit sphere – and this insight is precisely the reason that motivated [15] to consider a multi-scale approximation that can establish the $m \asymp w^2(T)/\epsilon^2$ guarantee for wider classes of random ensembles beyond the Gaussian case in Theorem 1. To put things in perspective with later sections, we work out the sample complexity required from a naive uniform discretization below.

**Lemma 2** ($\Delta$-net Covering). *Using Tensor-SRHT, with a uniformly constructed $\Delta$-net covering of the tensor, one requires $m = \mathcal{O}(\epsilon^{-2} \cdot n^{\frac{d^2}{1+d}} (\sum_{i=1}^d \gamma_2^2(T^i))^{\frac{d}{1+d}})$ for (2) to hold.*

Even in the prosaic case of Gaussian process indexed by ellipsoid and/or $\ell_1$ ball, it is a well-known and disappointing fact that arguments based on union bound / Dudley integral don't give the optimal bound, whereas method based on generic chaining does [16], which we turn to next.

## 4 A Multi-scale Approach: Row-wise Tensored Embedding

One viable approach is to apply the result of [15] naively to $\text{vec}(u^1 \otimes \cdots \otimes u^d)$ without taking into consideration the Kronecker structure, but this is somewhat of a futile endeavor if one takes any interest in downstream applications of such bounds. In fact, this was also the impetus for Mendelson's work on product empirical processes [14] – it is generally hard to handle geometric properties of process indexed by product classes. We will instead derive results with an eye towards bounds involving *decoupled* geometric complexity measure for each factor that lends itself to explicit computations – this necessarily calls for a more intricate chaining argument. Another possibility is to use a contraction inequality à la Ledoux-Talagrand if the random factors $\{v_i^j\}_{j=1}^d$ come from bounded class but this will be crude in almost all cases.

Our agenda is to leverage the results on finite set embedding from the previous section, wrap them inside of a chaining argument by exploiting coverings at multiple scales with different distortions/probability tradeoff so each level of approximation demands roughly the same embedding dimension (as we will see, the final $m$ depends on the maximum required across all resolutions).

### 4.1 Preliminaries

Throughout the paper, we use $\lesssim, \asymp, \gtrsim$ to hide absolute constants. To measure the size of the set $T^i \subset \mathbb{R}^n$, we use Gaussian width defined as for $g \sim \mathcal{N}(0, I_n)$,

$$ w(T^i) = \mathbb{E}\left[\sup_{u \in T^i} g^\top u\right]. $$

In our context, we define the $\gamma_2^*$ functional as

$$ \gamma_2^*(T^i) := \inf_{\{T_l^i\}} \sup_{u^i \in T^i} \sum_{l=0}^\infty 2^{l/2}\text{dist}(u^i, T_l^i) $$

where the infimum is taken over all sequences of nets $\{T_l^i\}_l$ with cardinality $|T_l^i| \leq 2^{2^l} =: N_l \ \forall i \in [d]$ and $|T_0^i| = 1 =: N_0$. For Gaussian process with canonical metric (i.e., Euclidean norm) on $T^i$, the expected supremum is completely characterized by $\gamma_2^*(T)$, i.e.,

$$\gamma_2^*(T^i) \asymp w(T^i)$$

where the upper bound is due to Fernique and the (much deeper, specific-to-gaussian-process) lower bound is due to Talagrand's majorizing measure theorem. A more general definition working with admissible sequences defines

$$\gamma_2(T^i) := \inf_{\{\mathcal{A}_l^i\}} \sup_{u^i \in T^i} \sum_{l=0}^{\infty} 2^{l/2} \mathrm{diam}(\mathcal{A}_l^i(u^i))$$

where the infimum is taken over all admissible sequences (i.e., increasing sequence of partitions of $T^i$ with $|\mathcal{A}_l^i| \leq N_l$ for all $l \geq 0$) and $\mathcal{A}_l^i(u^i)$ denotes the (unique) element of $\mathcal{A}_l^i$ that contains $u^i$. It is not hard to see that by picking one point arbitrarily from each element of the partition, one can build a net which implies that we always have $\gamma_2^*(T^i) \leq \gamma_2(T^i)$. In fact, the work of [20] shows that these two quantities are always of the same order.

It is also an immediate consequence that for an optimal admissible sequence $\{\bar{\mathcal{A}}_l^i\}_l$, picking $\{\bar{T}_l^i\}_l$ as a sequence of nets with cardinally $|\bar{T}_l^i| \leq N_l$ constructed by choosing the center point in every element of the partition set $\{\bar{\mathcal{A}}_l^i\}_l$, we have for all $u^i \in T^i$, $i \in [d]$,

$$\sum_{l=0}^{\infty} 2^{l/2} \mathrm{dist}(u^i, \bar{T}_l^i) \leq \inf_{\{\mathcal{A}_l^i\}} \sup_{t \in T^i} \sum_{l=0}^{\infty} 2^{l/2} \mathrm{diam}(\mathcal{A}_l^i(t)). \tag{3}$$

For our results, we will find it helpful to adopt the slightly more general $\gamma_\alpha$-functional for $\alpha > 0$:

$$\sum_{l=0}^{\infty} 2^{l/\alpha} \mathrm{dist}(u^i, \bar{T}_l^i) \leq \gamma_\alpha(T^i) := \inf_{\{\mathcal{A}_l^i\}} \sup_{u^i \in T^i} \sum_{l=0}^{\infty} 2^{l/\alpha} \mathrm{diam}(\mathcal{A}_l^i(u^i))$$

and the infimum is taken over all admissible sequences in exactly the same way as (3). It is known that for a random variable with tail decay bounded as $e^{-|x|^\alpha}$, the supremum is upper bounded by the $\gamma_\alpha$ functional [9]. Moreover, we always have the following Dudley-style metric entropy integral estimate [16] where $B_2^n$ denotes the unit-$\ell_2$ ball in $\mathbb{R}^n$:

$$\gamma_\alpha(T^i) \lesssim C_\alpha \int_0^1 \left(\log N(T^i, sB_2^n)\right)^{1/\alpha} ds, \tag{4}$$

but the reverse is generally not true. Here the upper limit of the integral goes up to 1 because $N(T^i, sB_2^n) = 1$ for $s \geq 1$ by simply picking $\{0\}$ as cover. Covering number on the RHS of (4) can be bounded with estimates on Gaussian width. In particular, Sudakov minorization asserts

$$\sup_{s>0} s\sqrt{\log N(T^i, sB_2^n)} \lesssim w(T^i),$$

which uses covering number at a single scale. Various alternative options exist for upper bounding the covering number, including Volumetric estimates, Maurey's empirical method etc.

Estimate (4) above has the drawback of not being explicit in constants $C_\alpha$, if one is keen on explicit dependence on $\alpha$, the following lemma becomes timely.

**Lemma 3** (Relationship between $\gamma_\alpha$ functionals). *For $\alpha \leq 1$, if set $T^i \subset \mathbb{S}^{n-1}$ has covering number $N(T^i, sB_2^n) \leq (\frac{a}{s})^b$ for some $b \geq 2$, $a \geq 2$, then*

$$\gamma_2(T^i) \leq \gamma_\alpha(T^i) \leq \left(1 + K \cdot \log_2(b/\alpha) \cdot b/\alpha \cdot \log_2(a)\right)^{\frac{2-\alpha}{2\alpha}} \gamma_2(T^i)$$

*for some absolute constant $K$.*

## 4.2 Multi-resolution embedding property

Instead of going through the multi-scale RIP (followed by column sign randomization) as done in [15] we will give ourselves more wiggle room by working with a multi-scale embedding property for finite sets. Definition 3 below will be featured prominently in subsequent sections and make the successive construction of approximations less mysterious than it may otherwise seem. We will invoke it for Tensor-SRHT and Tensor-Subgaussian in this section – both taking the form where each row $S_i = \mathrm{vec}(v_i^1 \otimes \cdots \otimes v_i^d)$.

**Definition 3** (Multi-resolution Embedding Property). *A mapping $S : \mathbb{R}^{n^d} \mapsto \mathbb{R}^m$ fulfills the $(\epsilon, \eta, \alpha)$-Multi-resolution Embedding Property if for an increasing sequence of successive coverings $\{\bar{T}_l^i\}_l$ of $T^i \subset \mathbb{S}^{n-1}$ such that $|\bar{T}_l^i| \leq 2^{2^l}$ and $|\bar{T}_0^i| = 1 \,\forall i \in [d]$ defined in (3) for tensor $x := u^1 \otimes \cdots \otimes u^d$, the following holds simultaneously for all $1 \leq l \leq L \asymp \lceil \log_2(nd) \rceil$ with probability at least $1 - \exp(-\eta)$:*

- *For all $k \in [d]$ and $l \in [L]$,*

$$
\big| \|S(u_l^1 \otimes \cdots \otimes u_l^k \otimes \cdots \otimes u_{l-1}^d) - S(u_l^1 \otimes \cdots \otimes u_{l-1}^k \otimes \cdots \otimes u_{l-1}^d)\|_2^2
$$
$$
- \|u_l^1 \otimes \cdots \otimes (u_l^k - u_{l-1}^k) \otimes \cdots \otimes u_{l-1}^d\|_F^2 \big|
$$
$$
\leq \max(2^{l/\alpha}\epsilon, 2^{2l/\alpha}\epsilon^2) \cdot \|u_l^1\|_2^2 \cdots \|u_l^k - u_{l-1}^k\|_2^2 \cdots \|u_{l-1}^d\|_2^2
$$

- *For all $k \in [d]$ and $l \in [L]$,*

$$
\big| \|S(u_l^1 \otimes \cdots \otimes u_l^k \otimes \cdots \otimes u_{l-1}^d)\|_2^2 - \|u_l^1 \otimes \cdots \otimes u_l^k \otimes \cdots \otimes u_{l-1}^d\|_F^2 \big|
$$
$$
\leq \max(2^{l/\alpha}\epsilon, 2^{2l/\alpha}\epsilon^2) \cdot \|u_l^1\|_2^2 \cdots \|u_l^k\|_2^2 \cdots \|u_{l-1}^d\|_2^2
$$

- *For all $k \in [d]$ and $l \in [L]$,*

$$
\left| \left\| S\left( u_l^1 \otimes \cdots \otimes \left( \frac{u_l^k - u_{l-1}^k}{\|u_l^k - u_{l-1}^k\|_2} \right) \otimes \cdots \otimes u_{l-1}^d \right) \pm S(u_l^1 \otimes \cdots \otimes u_{l-1}^k \otimes \cdots \otimes u_{l-1}^d) \right\|_2^2 \right.
$$
$$
\left. - \left\| u_l^1 \otimes \cdots \otimes \left( \frac{u_l^k - u_{l-1}^k}{\|u_l^k - u_{l-1}^k\|_2} \pm u_{l-1}^k \right) \otimes \cdots \otimes u_{l-1}^d \right\|_F^2 \right|
$$
$$
\leq \max(2^{l/\alpha}\epsilon, 2^{2l/\alpha}\epsilon^2) \cdot \left\| \frac{u_l^k - u_{l-1}^k}{\|u_l^k - u_{l-1}^k\|_2} \pm u_{l-1}^k \right\|_2^2 \cdot \|u_l^1\|_2^2 \cdots \|u_l^{k-1}\|_2^2 \|u_{l-1}^{k+1}\|_2^2 \cdots \|u_{l-1}^d\|_2^2
$$

*where tensor Frobenius norm $\|x\|_F := \prod_{k=1}^d \|u^k\|_2$ and $u_l^k$ is the closest point to $u^k$ in $\{\bar{T}_l^k\}$.*

For the desired accuracy $\epsilon > 0$ in the final guarantee (2), in what follows we correspondingly define a sequence of distortion levels $\epsilon_0 = \epsilon, \epsilon_1 = 2^{1/\alpha}\epsilon, \cdots, \epsilon_L = 2^{L/\alpha}\epsilon$ for $L \asymp \lceil \log_2(nd) \rceil$ levels and let $\tilde{L} = \max(0, \lfloor \alpha \log_2(1/\epsilon) \rfloor)$ such that for $l \leq \tilde{L}$, $\epsilon_l \leq 1$ therefore $\max(\epsilon_l, \epsilon_l^2) = \epsilon_l$. Additionally, we define $x = u_{L+1}^1 \otimes \cdots \otimes u_{L+1}^d$ being the finest level of approximation. Give $\epsilon, n, d$, we will pick $L = C\lceil \log_2(nd) \rceil$ for a constant $C$ and work under the assumption that $\tilde{L} \leq L$ in the proofs presented in Section B – the case when $\tilde{L} > L$ allows us to draw the same conclusion and is deferred to Appendix D. Here the constant $C$ is independent from all problem parameters.

Definition 3 takes center stage in the following lemma. The trade-off of $\eta_l$, $\epsilon_l$ and $p_l$ specified in the proof of Lemma 4 below ensures that there's no occurrence of $l$ in the final stated $m$. The $\{\epsilon_l\}$ plays the role of multi-level approximation close in spirit to what the $\gamma$-functional attempts to capture. The super-exponential factor of $d^d$ also made an appearance in earlier work on embedding of finite set using Tensor-SRHT [3].

**Lemma 4** (Multi-resolution embedding property of row-wise tensored sketches). *With $m = \mathcal{O}(C^d(d^d + (1+\eta)^d)/\epsilon^2)$, Tensor-SRHT defined in Definition 1 satisfies Definition 3 for $\alpha = 2/d$. The same property also holds for Tensor Subgaussian defined in Definition 2 for $m = \mathcal{O}(C^d\sigma^{2d}(d^d + (1+\eta)^d)/\epsilon^2)$ and $\alpha = 2/d$.*

### 4.3 Embedding of general sets with row-wise tensored sketches

Now we embark on our journey for the proof of our main result on row-wise Kronecker-structured sketches where Definition 3 and Lemma 4 will reveal their power.

**Theorem 2** (Gordon-type Inequality for Tensor-SRHT and Tensor-Subgaussian). *Tensor-SRHT with $m = \mathcal{O}(C^d\epsilon^{-2}(\sum_{i=1}^d \gamma_{2/d}(T^i))^2 d^d)$ satisfies uniform concentration (2). The same guarantee carries over to Tensor-Subgaussian with $m = \mathcal{O}(C^d\sigma^{2d}\epsilon^{-2}(\sum_{i=1}^d \gamma_{2/d}(T^i))^2 d^d)$.*

This recovers the result of [15] for $d = 1$ (ignoring poly-logs). In light of the tail bound Theorem 2.1 in [4], it is also natural that the $\gamma_{2/d}$ functional shows up.

*Remark.* This concentration result can also be easily converted to be on $|\|Sx\|_2 - 1|$ using basic inequality $\frac{1}{3}\min\{|a^2 - 1|, \sqrt{|a^2 - 1|}\} \leq |a - 1| \leq \min\{|a^2 - 1|, \sqrt{|a^2 - 1|}\}$ for $a \geq 0$. For a short proof, see Appendix B.

It is worth noting that the above argument will generalize to other structured random ensembles, e.g., partial circulant matrix with random signs. To put things in context, we compare this bound with what we got from Lemma 2. Using Lemma 3,

$$\gamma_{2/d}(T^i) \leq (1 + K \cdot \log_2(b/\alpha) \cdot b/\alpha \cdot \log_2(a))^{\frac{d-1}{2}} \gamma_2(T^i),$$

which means substituting into Theorem 2, assuming for the sake of argument all the $T^i$ are the same, focusing on the dependence on $\epsilon$ and $\gamma_2$, this approach gives

$$m = \mathcal{O}((\sum_{i=1}^{d} \gamma_{2/d}(T^i))^2 \epsilon^{-2}) = \mathcal{O}\left((b\log_2(a))^{d-1} \cdot \gamma_2(T^i)^2 \epsilon^{-2}\right). \tag{5}$$

if ignoring poly-logs. In contrast to Lemma 2 where we used a single-scale discretization $m = \mathcal{O}(\epsilon^{-2} \cdot n^{\frac{d^2}{1+d}}(\gamma_2^2(T^i))^{\frac{d}{1+d}})$, Sudakov informs us

$$\sqrt{b\log(a)} \leq \sup_{\epsilon \in (0,1]} \epsilon\sqrt{b\log(a/\epsilon)} \lesssim \gamma_2(T^i) \leq \sqrt{n}.$$

Therefore in the case of low complexity set ($\gamma_2(T^i) \ll \sqrt{n}$), the multi-resolution approach pays off.

## 5 Recursive Kronecker Embedding

The row-wise-tensored mapping from the previous section, despite its simplicity, gives exponential dependency on the degree $d$ (and necessarily so, as a preview for Section F), suggesting it is ideal for embedding low-degree tensor. In this section, we analyze the "sketch and reduce" approach proposed by [2], which composes degree-2 sketches from the previous section in the following way: we define the operation $S$ acting on rank-1 e.g., degree-3 tensor as

$$S(x \otimes y \otimes z) := S^1(x \otimes S^2(y \otimes S^3 z)). \tag{6}$$

The distinctive feature of the design is that at each layer, the Kronecker-structured sketch $S^k$ only acts on degree-2, reduced-dimensional tensor – something it excels at. It is an easy exercise that the matrix $S \in \mathbb{R}^{m \times n^d}$, when acting on rank-1 degree-$d$ tensor, can be deemed as $S = Q^0$ for

$$Q^d = 1 \text{ and } Q^{k-1} = S^k(Q^k \otimes I_n) \in \mathbb{R}^{m \times n^{d-k+1}} \text{ for } k = d, \cdots, 1,$$

where each $S^k \in \mathbb{R}^{m \times nm}$ for $k \in [d-1]$ and $S^d \in \mathbb{R}^{m \times n}$.

### 5.1 Building blocks for multi-resolution covering

The analysis follows the same template once we know how the JL moment property is preserved under matrix direct sum and multiplication, which was investigated in previous work. We have the following discrete JL property for the embedding matrix $S$ introduced above.

**Lemma 5** (Finite Set Embedding Property). *The recursive embedding* (6) *satisfies* $|\|Sx\|_2^2 - 1| \leq \max(\epsilon, \epsilon^2)$ *for all unit-norm, rank-1 tensors* $x \in \mathbb{R}^{n^d}$ *belonging to a finite set of cardinality $p$ with probability at least $1 - e^{-\eta}$ for any $\eta > 0$ with $m = \mathcal{O}\left(\frac{d}{\epsilon^2}(\log^2(p) + \eta^2 \vee \eta)\right)$. Moreover, such operation can be conducted in time $\mathcal{O}(d(n\log n + m))$ when each $S^i$ is constructed from an degree-2 Tensor-SRHT sketch.*

The ensuing lemma makes it clear that we should be grateful for the result stated above.

**Lemma 6** (Multi-resolution embedding property of Recursive Tensor-SRHT). *With $m = \mathcal{O}(d(d^2 + (1+\eta)^2)/\epsilon^2)$, Recursive Tensor-SRHT satisfies the $(\epsilon, \eta, \alpha)$-Multi-resolution Embedding Property in Definition 3 with $\alpha = 1$.*

## 5.2 Embedding of general set using recursive sketch

We will employ a slightly different decomposition of the chain for this construction and dedicate the section to prove the following theorem. At a high level, the observation is that the sketch, albeit taking complicated hierarchical form, happens to be linear when acting on rank-1 tensor. Therefore the strategy is to have all the terms in the chain we need to control in the rank-1 form that only involves difference in one factor, after which the multi-resolution embedding property can be repeatedly instantiated as before.

**Theorem 3** (Gordon-type Inequality for Recursive Kronecker Embedding). *The Recursive Tensor-SRHT with $m = \mathcal{O}(d\epsilon^{-2}(\sum_{i=1}^{d} \gamma_1(T^i))^2 \cdot (d^2 + (1+\eta)^2))$ satisfies $|\|Sx\|_2^2 - 1| \leq \max(\epsilon, \epsilon^2)$ for all $x = u^1 \otimes \cdots \otimes u^d \in T^1 \times \cdots \times T^d$ with probability at least $1 - \exp(-\eta)$ for $d \geq 2$.*

It is enlightening to compare with the previous embedding bound, assuming again the covering number admits $N(T^i, sB_2^n) \leq (\frac{a}{s})^b$ for all $i \in [d]$. With (4) we have

$$\gamma_1(T^i) \leq C_1 \int_0^1 \log N(T^i, sB_2^n)\, ds \leq C_1 \int_0^1 b \log(a/s)\, ds \leq C_1' \cdot b \log(a)$$

which means using Theorem 3 that $m = \mathcal{O}(d^5 b^2 \log^2(a)/\epsilon^2)$ for the desired embedding guarantee. This is favorable as the dependence on $d$ has been reduced from exponential to polynomial. For example we can see that when each $T^i$ consists of a set of $p$ points on the unit sphere, $b = o(1)$ and $a = p$ we get $\log^2(p)/\epsilon^2$ as opposed to $\log^d(p)/\epsilon^2$ from the previous section (5) when focusing on the scaling with $p$.

# 6 Applications

In this section, we give applications of our result in two settings, deploying one type of random embedding for each, where we see how these bounds can take advantage of the underlying low complexity structure to move away from the (potentially much larger) ambient dimension. We note that these applications crucially exploit the fact that the object in $\mathbb{R}^{n^d}$ being acted upon has Kronecker structure – this departs from e.g., oblivious subspace embedding (OSE) result from [1] where the column span of *any* $n^d \times p$ matrix is preserved.

## 6.1 Signal Recovery

Inspired by compressed sensing, suppose we are given independent random (linear) 1-subgaussian measurements on Kronecker-structured rank-1 signal $x$ of type

$$y_i = \langle S_i, x \rangle = \prod_{j=1}^{d} \langle v_i^j, u^j \rangle,\ i \in [m] \tag{7}$$

for $x = u^1 \otimes \cdots \otimes u^d$, $u^i \in T^i \subset \mathbb{S}^{n-1}$, and would like to know when does performing

$$\min_{\{z^j\}_{j=1}^d \in \mathbb{S}^{n-1}} \sum_{j=1}^{d} f_j(z^j) \quad \text{subject to } S(z^1 \otimes \cdots \otimes z^d) = y,\ f_j(z^j) \leq R_j\ \forall j \in [d] \tag{8}$$

uniquely reconstruct $x$, where $f_j$ above is convex and $R_j := f_j(u^j)$ encodes the prior knowledge we have so that $\{u^j\}$ is feasible. In the case when such information is not available, the constraint can simply read as $\|z^j\|_2 \leq 1$, for example. Notice that the decision variable lives in a lower dimensional space ($nd$ as opposed to $n^d$ if we naively vectorize the signal) and one candidate could be alternating projected gradient descent over each factor. Computation aside on which algorithm to enlist for solving (8) (it involves solving a non-convex problem due to the multi-linear structure), the analysis below gives an information-theoretic lower bound on the sample complexity for successful recovery. The following quantities facilitate the analysis.

**Definition 4** (Descent Cone and Restricted Singular Value). *We use $\mathcal{D}(f_j, u^j)$ to denote the descent cone of a convex function $f_j$ at point $u^j \in \mathbb{R}^n$, that is, $\mathcal{D}(f_j, u^j) := \cup_{\tau > 0}\{t \in \mathbb{R}^n : f_j(u^j + \tau t) \leq f_j(u^j)\}$. The correspondingly normalized descent cone is denoted as $\bar{\mathcal{D}}(f_j, u^j) := \mathcal{D}(f_j, u^j) \cap \mathbb{S}^{n-1}$. Let $\sigma_{\min}(S; \mathcal{C})$ be the minimum singular value of a matrix $S$ restricted to set $\mathcal{C}$, i.e., $\sigma_{\min}(S; \mathcal{C}) := \min_{x \in \mathcal{C} \cap \mathbb{S}^{n-1}} \|Sx\|$. Furthermore, the descent cone of a proper convex function is always convex.*

We take hints from [8, 18] for the lemma below.

**Lemma 7** (Recovery Guarantee). *If $\|Sw\| \geq (1-\epsilon)\|w\|$ for all $w = (u^1+t^1) \cap \mathbb{S}^{n-1} \otimes \cdots \otimes (u^d + t^d) \cap \mathbb{S}^{n-1}$ for which $t^j \in \mathcal{D}(f_j, u^j)$ where $\epsilon < 1$, the optimizer $\{z_*^j\}_{j=1}^d$ returned by (8) satisfies $z_*^1 \otimes \cdots \otimes z_*^d = u^1 \otimes \cdots \otimes u^d$ for the measurement model (7).*

Invoking Theorem 2 with Tensor-Subgaussian, for $\epsilon \in (0,1)$, $\forall w \in \mathcal{W}^1 \times \cdots \times \mathcal{W}^d$ where $\mathcal{W}^j := (u^j + \mathcal{D}(f_j, u^j)) \cap \mathbb{S}^{n-1}$,

$$|\|Sw\| - 1| \leq \min\{|\|Sw\|_2^2 - 1|, |\|Sw\|_2^2 - 1|^{1/2}\} \leq \epsilon$$

if picking $m = \mathcal{O}(C^d(\sum_{i=1}^d \gamma_{2/d}(\mathcal{W}^i))^2 \cdot (d^d + (1+\eta)^d)/\epsilon^2)$, which means $\sigma_{\min}(S; \mathcal{W}^1 \times \cdots \times \mathcal{W}^d) \geq 1 - \epsilon > 0$ as needed by Lemma 7.

Using translation-invariance and subadditivity of the $\gamma$-functionals, one could show that this is order-wise the same as $m = \mathcal{O}(C^d(\sum_{i=1}^d \gamma_{2/d}(\bar{\mathcal{D}}(f_i, u^i)))^2 \cdot (d^d + (1+\eta)^d))$ if $f_j$'s are convex – we refer to Appendix E for details. Now thanks to the decoupling, it reduces to $d$ descent cone vector Gaussian width type calculation.

**Example 1.** *Supppose each of the $d$ factors is $k$-sparse, i.e., $T^i = \{u^i \in \mathbb{R}^n : \|u^i\|_0 \leq k, \|u^i\|_2 = 1\}$, it is classical that the normalized descent cone for $\ell_1$ norm at $k$-sparse vector is $\bar{\mathcal{D}}(f_i, u^i) = \{s : \|s\|_1 \leq 2\sqrt{k}\|s\|_2, \|s\|_2 = 1\}$. Since $conv(kB_0^n \cap B_2^n) \subset \sqrt{k}B_1^n \cap B_2^n \subset C \cdot conv(kB_0^n \cap B_2^n)$ for an absolute constant $C$, from known result one can deduce that the covering number and Gaussian width scale as*

$$w(\bar{\mathcal{D}}(\|\cdot\|_1, u^j)) \asymp \sqrt{k \log(en/k)}$$
$$\log(|\mathcal{N}^{\Delta}(\bar{\mathcal{D}}(\|\cdot\|_1, u^j))|) \asymp k\log(en/\Delta k),$$

*consequently*

$$\gamma_{2/d}^2(\mathcal{D}(\|\cdot\|_1, u^j)) \lesssim (kd\log(n/k)\log(kd))^{d-1} \cdot k\log(n/k).$$

*This gives assuming $\log(n/k) \ll k$ (not worrying about the $d^d$ factor, assuming $d$ is small for this application) with $m = \mathcal{O}\left(k^d(1+\eta)^d\right)$, the recovery is successful with probability at least $1 - \exp(-\eta)$ when omitting poly-logs. It should be clarified that the minimizer of (8) may not be unique (as in the case with $f_j = \|\cdot\|_1$ up to sign ambiguity – which is the only possible one for rank-1 tensor), but this sample complexity suffices for recovering any of the equivalent representations of the rank-1 signal under consideration.*

In general, the work of [8, 18] provide powerful recipe for bounding the Gaussian width of a descent cone based on duality and polar cones: for $f_j$ a convex function, and $u^j \in \mathbb{R}^n$ a fixed point, $g \sim \mathcal{N}(0, I_n)$,

$$w^2(\mathcal{D}(f_j, u^j)) \leq \mathbb{E}\inf_{\tau \geq 0} \text{dist}^2(g, \tau \cdot \partial f_j(u^j)),$$

which cries out for more applications for such structured tensor recovery problems.

## 6.2 Optimization

Consider an optimization (tensor decomposition) problem, where for given signal $x = u^1 \otimes \cdots \otimes u^d \in T^1 \times \cdots \times T^d$ taking Kronecker structure, we wish to solve for

$$\min_{z^i \in T^i \, \forall i \in [d]} \|u^1 \otimes \cdots \otimes u^d - z^1 \otimes \cdots \otimes z^d\|_F^2.$$  (9)

In general, one could also consider the denoising version where there is noise in the observation $x + e$, but for simplicity we focus on the noiseless case below. With the hope of saving storage and speeding up, we apply sketching before solving a lower $m$-dimensional problem:

$$\min_{z^i \in T^i \, \forall i \in [d]} \|S(u^1 \otimes \cdots \otimes u^d) - S(z^1 \otimes \cdots \otimes z^d)\|_2^2 =: g(z^1, \cdots, z^d).$$  (10)

Let $S$ be the recursive sketch from Section 5 and denote the optimizer of (10) as $\{z_*^i\}$. It is not hard to see that since $g(z_*^1, \cdots, z_*^d) \leq g(u^1, \cdots, u^d) = 0$, we must have $S(z_*^1 \otimes \cdots \otimes z_*^d) = S(u^1 \otimes \cdots \otimes u^d)$, which means that $S$ restricted to set $T^1 \times \cdots \times T^d$ must have the smallest singular value bounded away from 0 for us to uniquely identify the rank-1 factors. Note again this doesn't resolve the inherent ambiguity between the factors such as sign flips but the resulting sample complexity is sufficient to recover any such signal consistent with the measurement (i.e., the returned rank-1 solution obeys $z_*^1 \otimes \cdots \otimes z_*^d = u^1 \otimes \cdots \otimes u^d$ hence in $x$ space it is unique). We give an example in Section E.

# 7 Experiments

In this section, we numerically investigate (1) embedding dimension scaling with $d$ for the two types of random embeddings in Section 4 and 5; (2) signal recovery from random Gaussian measurements as elaborated in Section 6.1 where the signal is rank-1 belonging to a product of cones. This is in some sense a generalization of the non-convex Wirtinger flow formulation for the phase retrieval problem [7], where both the random measurement and the signal are non-symmetric, in addition to the availability of potential prior knowledge on the factors.

For the first experiment, we let $n = 10, d = 5$ and pick each factor $\{u^j\}$ to be 20% sparse. The figure below reports the average distortion of the embedding $|\|Sx\|_2 - 1|$ over 25 runs for both the row-wise tensored and recursive sketch with Gaussian random factors. Variance of the distortion across the trials is also shown as we vary $m = 0.8 \times n \times d^\alpha$ for $\alpha \in \{1, \cdots, 5\}$. For the second experiment, we perform projected gradient descent on the following objective:

$$\min_{\|z^1\|_1 \leq R_1, \cdots, \|z^d\|_1 \leq R_d} \frac{1}{2m} \sum_{i=1}^m \left( y_i - \prod_{j=1}^d \langle v_i^j, z^j \rangle \right)^2 =: \mathcal{L}(z^1, \cdots, z^d). \tag{11}$$

We use the spectral factorization of $\frac{1}{m} \sum_{i=1}^m \langle v_i^1, u^1 \rangle v_i^1 \otimes \cdots \otimes \langle v_i^d, u^d \rangle v_i^d = \frac{1}{m} \sum_{i=1}^m y_i v_i^1 \otimes \cdots \otimes v_i^d$ as initialization, as in expectation this is the signal $x = u^1 \otimes \cdots \otimes u^d$ since we assumed $v_i^j$ are independent across the $j$'s. We use the `tucker-als` function from the Matlab Tensor Toolbox[1] for computing the best rank-$(1, 1, 1)$ tensor approximation, after which gradient update is made on each factor followed by $\ell_1$ projection. We set each factor $\{u^j\}$ to be 20% sparse and let $d = 3, n = 10$, $m = 2 \times 0.8 \times n \times d^\alpha$ for $\alpha \in \{1, \cdots, 3\}$ and record the successful recovery out of 25 trials. Stepsize is picked to be 0.1 and success is defined as $\mathcal{L}(z^1, \cdots, z^d) \leq 0.1$ after 500 gradient steps.

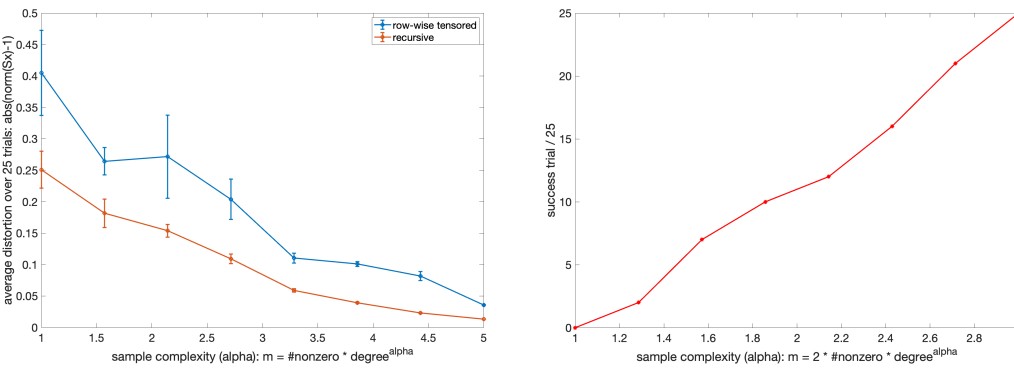

Figure 1: Left: Embedding scaling with degree. Right: Tensor recovery from Gaussian measurement.

# 8 Discussion

In this paper, we generalized Gordon's inequality to two families of tensor-structured random embeddings, which admit efficient implementation when acting on rank-1 Kronecker-structured signals. As future work, we deem rigorously establishing a lower bound on the embedding dimension for general sets both an interesting and challenging direction to pursue. On the other hand, tools developed here should be helpful for analyzing performance of the gradient-based recovery algorithms that we numerically tested in Section 7.

## Acknowledgments and Disclosure of Funding

Our sincere appreciation goes to Rachel Ward for several discussions related to the topic. This work was performed at UT Austin supported under NSF IFML 2019844 and NSF 1934932.

---

[1]http://www.tensortoolbox.org

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
