# OpenReview forum: "Near-Isometric Properties of Kronecker-Structured Random Tensor Embeddings"
_NeurIPS.cc/2022/Conference — NeurIPS 2022 Accept_

### Official Review · Reviewer_vzAY · 2022-07-10

**Rating:** 5
**Confidence:** 4
**Soundness:** 3 good
**Presentation:** 3 good
**Contribution:** 2 fair

**Summary:**

The paper studies the concentration inequality for random tensor acting on rank-1 Kronecker structured signals and derives a similar near isometric properties for Kronecker structured random tensor embeddings. Two applications related to compressive sensing and tensor decomposition are analyzed. The proposed theory is tested in numerical experiments. The paper is well organized and written clearly.

**Questions:**

1. The reviewer is not clear what's the main innovation compared with the prior work [1]. This is the major question of the reviewer.
[1] Jin, Ruhui, Tamara G. Kolda, and Rachel Ward. "Faster johnson–lindenstrauss transforms via Kronecker products." Information and Inference: A Journal of the IMA 10.4 (2021): 1533-1562.

2. Due to the nonconvexity nature of the Kronecker structures, what other potential applications instead of those inverse problems discussed in the paper?

**Limitations:**

1. The major contribution of this work is not clear. The main idea of generalizing of the RIP to Kronecker structures has already been extensively explored in the literature. For example, in [1], the authors introduced a generalization of the fast Johnson-Lindenstrauss projection for embedding vectors with Kronecker product structure, the Kronecker fast JohnsonLindenstrauss transform (KFJLT), which is essentially the same topic considered in this work. The reviewer is not clear what's the main innovation compared with the prior works.

2. The proposed applications are interesting, but have some practical issues. Due to the nonconvexity, it is not that meaningful to study the properties of its global optimal solutions.

3. The experimental evaluation of the proposed methods is limited.

**Strengths And Weaknesses:**

*Strengths*
1. Derive a generalization of the Restricted Isometry Property (RIP) to higher-order tensored signals. Also, the proposed Kronecker structured random tensor embeddings allow for efficient implementations (e.g., recursive Kronecker embedding).
2. The paper is well organized and written clearly.
3. The potential applications are interesting, for example, the compressive sensing of tensor structured signal and tensor decomposition problems.

*Weaknesses*
1. The major contribution of this work is not clear. The main idea of generalizing of the RIP to Kronecker structures has already been extensively explored in the literature. For example, in [1], the authors introduced a generalization of the fast Johnson-Lindenstrauss projection for embedding vectors with Kronecker product structure, the Kronecker fast JohnsonLindenstrauss transform (KFJLT), which is essentially the same topic considered in this work. The reviewer is not clear what's the main innovation compared with the prior works.

[1] Jin, Ruhui, Tamara G. Kolda, and Rachel Ward. "Faster johnson–lindenstrauss transforms via Kronecker products." Information and Inference: A Journal of the IMA 10.4 (2021): 1533-1562.

2. The proposed applications are interesting, but have some practical issues. More precisely, the authors show that under some assumptions the global optimal solution coincides with the ground truth. However, the underlying optimization problems of both applications are essentially nonconvex due to the multi-linear structures. Unlike traditional compressive sensing, the underlying optimization problem is convex, where the global optimal solution is possible/easy to achieve. Due to the nonconvexity, it is not that meaningful to study the properties of its global optimal solutions.

3. The experimental evaluation of the proposed methods is limited.

---

> ### Author Response · Authors · 2022-08-01
> **Response #3**
>
> - Related Work: Thank you for bringing up the work of [JKW ‘21]. There are a few differences between the questions addressed in this work and the above-mentioned paper which we elaborate below: (1) The main result of Theorem 2.1 in the reference is a JL result, which means it holds for embedding finite point set only. Corollary 2.1 in the referenced paper for CP tensor decomposition application leverages a single-scale eps-net covering for extending the result to a particular example of continuous set embedding (i.e., column space). It is, however, known that for general cones, this construction would yield a suboptimal embedding dimension, even when d=2 (c.f. [19]), which is something we aim to address in our work. In terms of the technique used, the work of [JKW ‘21] leverages the well-known reduction of RIP to JL for random matrix when d=2 using sign randomization, and inducted the result on the degree for tensors (c.f. Proof of Proposition 4.2). In particular, the argument does not establish RIP properties for the embedding $SUD_\zeta$; the $SU$ in eqn (16) in the reference is an RIP matrix but all one could say about the final embedding $SUD_\zeta$ with the argument given therein is the JL lemma; (2) The result of [JKW ‘21] holds for arbitrary vectors, and not the Kronecker-structured signal as we consider. This was, for us, one main motivation for leveraging a correspondingly tensor-structured embedding, since this embedding operation can be performed efficiently when acting on rank-1 tensor-structured signals; (3) We additionally consider a recursive embedding operation that significantly improves the sample complexity dependence on d. We will clarify and highlight these distinctions in the revised version.
>
> - Applications and nonconvexity: Thanks for the comment. The non-convexity and non-uniqueness of the global solution is reminiscent of the phase retrieval problem (with the corresponding non-convex Wirtinger flow formulation), in which case d=2 and the recovered signal is only unique up to signs. Under the rank-1 normalized model that we consider for tensors in this work, it is also the case that the only ambiguities are in the signs of the factors, which nevertheless yield the same recovered tensor and is the object of interest after all. In our experiments (and this is also similar to the phase retrieval/other non-convex formulation of signal recovery problems), we initialized the algorithm with a spectral factorization, followed by simple gradient updates. With the sample complexity that scales with the “complexity” of the signal, one is able to successfully recover the structured rank-1 tensor, with the factors usually converging to the one closet to where one initializes.

---

### Official Review · Reviewer_616B · 2022-07-11

**Rating:** 8
**Confidence:** 4
**Soundness:** 4 excellent
**Presentation:** 3 good
**Contribution:** 4 excellent

**Summary:**

The paper analyzes a class of sketchings that write as tensor products of vectors (i.e. random rank one tensor or Kronecker product of vectors if flattened). We know that the sketching discussed here can be calculated for a considerably lower computational cost as a non-structured sketching (to recover unstructured signal). The paper proves that the sketch (also referred to as random embedding) has statistical properties that can be written using properties of each individual signal component (Gaussian width of each of the Cones T1, ... Td): this a strong result because it is not a function of the joint variable (i.e. cartesian product T1 x T2 x .. x Td): this makes it considerably easier to get estimate bounds for the jointly structured sketches.
The paper discusses (Section 4) a class of random sketches which can be modeled as multi-scale embeddings. It is proven that the minimum embedding dimension m required scales at least exponentially with the tensor order / degree d.
The paper then analyzes a recursive embedding scheme in Section 5 where the paper's Th. 3 result interestingly drops the dependence on the degree (or order of Tensor) d to a polynomial factor.
The paper provides two applications / examples of Signal Recovery and Optimization problems that highlight the power of their results.

**Questions:**

is the Th. 3 result tight? the dependence in d, despite being poly, still feels a bit strong to me. Do you have any numerical experiment that at least suggest that the bound is tight? or do you have intuitions why it can be improved?

**Limitations:**

Examples in Section 6 are great, but I suspect that a more elaborate discussion on applications could improve the paper's impact.

**Strengths And Weaknesses:**

Strengths: The paper proves that the sketch has statistical properties that can be written using properties of each individual signal component (Gaussian width of each of the Cones T1, ... Td): this a strong result because it is not a function of the joint variable (i.e. cartesian product T1 x T2 x .. x Td).
The paper's Th. 3 result is very interesting in that it drops the dependence on the degree (or order of Tensor) d to a polynomial factor: there is a statistical and computational reduction in using tensor structured sketches provided that the signal contains this structure. This is a strong result.
The paper does a good job citing recent papers that have established analogous results in related problems. The paper's Applications examples in Section 6 are concise and convincing.

Weaknesses: I am not able to flag any technical weaknesses in the paper.
The paper's presentation is a bit rough. It can take time to the reader to find definitions of the objects introduced (sometimes they're defined a few lines before or after or in cited papers). Example: Definition 1 introduces lots of objects including P_i's that are not defined. I guess they contain a single 1 at one row. Other example: Eq. (2) requires some parsing to figure out what is randomized (it's v's) and epsilon pops out of nowhere. Another notation choice that bothered me is the use of the same letter "a" for  a vector and a tensor at lines 113 & 114.

---

> ### Author Response · Authors · 2022-08-01
> **Response #2**
>
> We appreciate all the comments and feedback.
>
> - Notation and Definition: Yes $P_i$’s are random subsampling matrices with a single 1 in each row in Definition 1. In eqn (2), the $\{v_i^j\}$’s are random and $\epsilon$ refers to the desired embedding distortion. Thanks for the suggestions we will clarify the notation /  add a discussion section in the revised version.
> - Tightness: We conducted an experiment that numerically tested the scaling distortion with the power of d (c.f. The left plot of Figure 1 in Appendix G). While it is still a small-scale experiment, the improvement of the recursive sketch over the row-wise tensored sketch on the dependence of d is clear. Rigorously establishing a lower bound on the embedding dimension for the recursive sketch for general sets, is, in our opinion, both an interesting and challenging future direction.

---

### Official Review · Reviewer_Udiw · 2022-07-12

**Rating:** 6
**Confidence:** 2
**Soundness:** 3 good
**Presentation:** 3 good
**Contribution:** 3 good

**Summary:**

This paper establishes a uniform concentration inequality for Kronecker-structured random rank-1 tensors, with a focus on the embedding dimension $m$. The authors consider two random tensor models and characterize the dependence of $m$ on the complexity of the signal $\gamma$ as well as the order $d$. In order to improve its dependence on $d$,  a recursive embedding is further proposed. Finally, the authors apply the general tool to two concrete applications to illustrate its effectiveness.

**Questions:**

1. It would make the paper more accessible to readers if the authors can summarize the notations in an individual section. For example, the authors should provide the definitions of Hadamard product and the transposed Khatri-Rao product in Definition 1.

2. In Definition 2, it seems that $\sigma^2$ is missing in (1) and that there is an extra square root in (2)?

**Strengths And Weaknesses:**

Strength:
Overall, the paper is well-written and easy to read. The results are interesting and novel, which would certainly contribute to the field.

Weakness:
1. The paper seems to be incomplete --- theoretical guarantees for the tensor decomposition problem in Section 6.2 is missing, and so does the conventional discussion and future direction section.
2. The discussion on the tightness seems insufficient, and part of it relies on a special case. It would improve the quality of the paper to establish the lower bounds for more general cases.
3. For the two applications considered in Section 6, it would be better to compare the results with existing works so as to show the advantage of the general tool developed in this paper.

---

> ### Author Response · Authors · 2022-08-01
> **Response #1**
>
> Thank you for the comments and suggestions.
>
> - Notation and Definition 2: Thanks we will address them in the revised version.
> - Tightness: The case for general non-Gaussian embedding with continuous set is notoriously hard to establish. It is known Gordon’s lemma is essentially tight for convex sets for Gaussian random matrix ([20]). For distributions beyond Gaussian, even in the case d=2 for continuous set, there still lack definitive answers on the tightness of the embedding dimension in general. There are, however, results for embedding particular sets: e.g., JL-lemma [A1] and subspace embedding [A2]. It is our intention to give a few concrete examples where our bound is tight for our tensor embedding in this paper, but we deem a lower bound for general sets interesting future direction to pursue.
> - Applications: To the best of our knowledge, we are not aware of prior work studying rank-1 tensor recovery with the type of embedding we consider. There are, however, various works studying random measurements taken with tensors of i.i.d Gaussian entries (e.g., [A3]). For the tensor decomposition/optimization, [A4] considered CP tensor decomposition, but crucially the requirement of oblivious subspace embedding property there requires the column span of any $n^d × p$ matrix is preserved, which is different from the setup we consider where we exploit the fact that the object acted upon in $\mathbb{R}^{n^d}$ has Kronecker structure.
> - Exposition: The guarantee for Sec 6.2 was mentioned in the text. We didn’t quite consider it a lemma but perhaps will turn it into an example with appropriate macros to highlight the result of the section. Due to the space constraints the “discussion and future work” part was left out in the main text but thanks for the comment we will include them in the revised version.
>
> [A1] Larsen, Nelson “Optimality of the Johnson-Lindenstrauss lemma”
>
> [A2] Tropp, “Improved Analysis of the Subsamples Randomized Hadamard Transform”
>
> [A3] Tong, Ma, Prater-Bennette, Tripp, Chi, “Scaling and Scalability: Provable Nonconvex Low-Rank Tensor Estimation from Incomplete Measurements”
>
> [A4] Jin, Kolda, Ward, “Faster johnson–lindenstrauss transforms via Kronecker products ”

---

### Meta-Review · Area_Chair_6yUA · 2022-08-25

**Recommendation:** Accept
**Confidence:** Certain

**Metareview:**

The paper derives uniform concentration inequalities for random tensors acting on rank-1, in the spirit of Gordon's inequality.
This type of results is at the root of a large literature based on the Convex Min-Max theorem which recently allowed for the analysis of plethora of empirical risk minimization problems in convex settings. The paper goes in the direction of providing the necessary tools to go beyond matrices and extend to projections of low rank signals on random tensors. Therefore such results have a very high potential in terms of applicability and is highly relevant to the ML community.
The reviewers are all favorable and I agree with them on the quality of the paper and its presentation. I also acknowledge the detailed answers provided during the rebuttal period. I'm confident that the authors will implement the relevant suggestions of the reviewers to fine tune the final version of the paper.

**Award:**

No

---

### Decision · Program_Chairs · 2022-09-14

Accept